# Synergistic Effects of Ionizing Radiation Process in the Integrated Coagulation–Sedimentation, Fenton Oxidation, and Biological Process for Treatment of Leachate Wastewater

Sha Liu [1,2], Arindam Sinharoy [1], Ga-Young Lee [1], Myun-Joo Lee [3], Byung-Cheol Lee [3] and Chong-Min Chung [1,*]

[1]  Department of Environment Science & Biotechnology, Jeonju University, Jeonju 55069, Republic of Korea; liusha@jj.ac.kr (S.L.); arindam.sinharoy004@gmail.com (A.S.); dlrkdud1007@jj.ac.kr (G.-Y.L.)
[2]  Department of Life Sciences, Hengshui University, Hengshui 053000, China
[3]  Uniscan, Co., Ltd., Seoul 05836, Republic of Korea; myunjoolee@hanmail.net (M.-J.L.); bclee@uniscan.co.kr (B.-C.L.)
[*]  Correspondence: cmchung@jj.ac.kr

**Abstract:** This study evaluated the feasibility of ionizing radiation combined with coagulation–sedimentation and Fenton oxidation as a treatment method for landfill leachate. The experiments revealed a positive correlation between pollutant removal efficiency and increased ionizing radiation intensity. Remarkable pollutant removal efficiencies were achieved under ionizing radiation at 50 kGy, with a maximum of 27% removal of total organic carbon (TOC), 61% removal of total nitrogen, 51% removal of total phosphorus, and an impressive 93% removal of $NO_3^-$-N. With the addition of coagulation–sedimentation and Fenton oxidation, the treatment efficiency further increased by 33% nitrogen, 18% SCOD, and 8% phosphate. The most significant observation from the study was that for all the different treatment methods, the results were always better for leachate samples treated with ionizing radiation than for the untreated samples. Subsequently, biological treatment was applied as a post-treatment method to remove residual organic carbon and nitrogen, which found that the best removal efficiencies were only for the low salt concentration (0.5%) and the removal decreased with increasing salt concentration. These experimental results conclusively demonstrated that when treating leachate wastewater, it was more appropriate to employ physicochemical methods rather than a biological treatment, primarily due to the high salt concentration present.

**Keywords:** ionizing radiation; coagulation–sedimentation; Fenton oxidation; biological treatment; leachate wastewater



## 1. Introduction

The most common method for municipal solid waste disposal is through landfills, which eventually create secondary pollution by generating leachate [1,2]. Landfill leachate is a foul-smelling black or yellowish-brown liquid comprising a large amount of organic and inorganic substances, including hard-to-degrade organic substances (such as aromatic compounds and humic substances), inorganic salts (such as ammonia, carbonate, and sulfate), and metal ions (such as chromium, lead, and copper) [3]. The primary features of waste leachate include (a) a high level of organic contaminants, with chemical oxygen demand (COD) levels reaching tens of thousands of milligrams per liter, (b) it includes a range of recalcitrant and toxic contaminants, including heavy metal ions and hazardous organic compounds, and (c) a high mass concentration of ammonia nitrogen, ranging from hundreds to tens of thousands of milligrams per liter, which severely inhibits and decreases microbial activity and limits the scope of its biological treatment [4–6]. Because landfill leachate can contaminate ground and surface water, it is a significant public health concern, and it can have a long-term consequence on the surrounding ecosystem.

Waste leachate treatment has been a challenging issue in the water treatment industry [7,8]. So far, several leachate treatment systems have been proposed including

physicochemical and biochemical methods [9]. The conventional biological treatment systems, i.e., both aerobic and anaerobic processes, are often ineffective in dealing with the difficult-to-degrade organic matter present in landfill leachate as these processes are primarily designed for easily degradable organic matters [6,10]. This is also supported by the fact that the biochemical oxygen demand ($BOD_5$) to COD ratio of landfill leachate is ≤0.4, which falls under the category of less biodegradable compounds [11]. Moreover, due to the excess chlorine ion concentration (8% to 11%) in leachate, it is difficult to treat using biological methods [12]. Another problem faced by the biological treatment system is the presence of a high concentration of ammonia nitrogen and inorganic (heavy metals) ions, which are inhibitory to the nitrifying microorganisms present in the wastewater treatment system [13]. Hence, to achieve a better treatment efficiency using biological methods, often a pre-treatment step involving a physicochemical process is necessary [14,15].

Among the physicochemical methods, coagulation is the simplest process to remove suspended particles from wastewater [16,17]. However, coagulation alone cannot eliminate the refractory organic contaminants from landfill leachate and requires additional treatment [1]. One such treatment method is the advanced oxidation process (AOP) which can effectively degrade recalcitrant pollutants. AOPs include ozonation [18], Fenton oxidation [19,20], photocatalytic oxidation, and electrocatalytic oxidation [21–23].

Ionizing radiation technology is another unique AOP-based wastewater treatment method that employs gamma rays or high-energy electron beams as radiation sources to eliminate hard-to-degrade pollutants [24,25]. Ionizing radiation can directly interact with the pollutants or can degrade pollutants by producing hydroxyl and other reactive species from water due to its exposure to high energy radiation [26,27]. Ionizing radiation technology was examined in this study as the primary treatment method to provide an effective treatment to the landfill leachate. Another AOP, Fenton oxidation was also used as an additional treatment for landfill leachate in this current study due to its operational simplicity and relative low cost [28,29]. In Fenton oxidation, ferrous iron is used as a catalyst and hydrogen peroxide as an oxidant, which under optimum conditions generates hydroxyl radicals (redox potential of 2.8 EV) that can rapidly decompose and mineralize organic pollutants present in the wastewater [30].

There are a number of previous studies focusing on AOP-based treatment of landfill leachate [28,31–33]. However, very few focused on a combination of different methods such as ionizing radiation, Fenton oxidation, and coagulation. In this study, these methods are explored together, along with biological methods, to provide a better treatment efficiency compared to the individual technologies. The objectives of this study include optimization of the dosage and operational conditions for ionizing radiation, Fenton oxidation, and coagulation processes to evaluate their combined treatment potential on landfill leachate. The potential of an activated sludge-based biological method as the post-treatment technology was also explored in this study.

## 2. Result and Discussions

### 2.1. Leachate Treatment Using Ionizing Radiation

Figure 1 shows the removal efficiencies of pollutants from landfill leachate at different irradiation absorption doses. The maximum pollutant removal of 20% TCOD and 60.9% TN was obtained with 10 kGy irradiation. Although, there are not many previous works focusing on landfill leachate treatment using ionizing radiation, the results obtained in this work are comparable with those observed in the limited previous studies available. For example, Bae et al. [34] reported a 20.7% removal of dissolved organic carbon using 15 kGy irradiation, which is relatable to the TOC removal value obtained in the current work. Another study reported a maximum of 45.3% COD removal from leachate at an irradiation dose of 4 kGy [2]. The low initial COD concentration of the leachate (950–983.3 mg $L^{-1}$) was the reason behind a better performance at low irradiation intensity compared to this study. The main objective of ionizing radiation treatment is not direct pollutant removal but rather to make the pollutants more susceptible to the subsequent treatment, including biological

treatment, as evidenced by the reduction in less-biodegradable humic substances by 33.6% in landfill leachate [34]. This improvement in treatment efficiency was not only reported for landfill leachate but also for other types of wastewaters such as in textile industry [35], papermill [36], and swine wastewater [37]. A study reported that biodegradability of textile and dye wastewater increased by as high as 224% due to ionizing radiation treatment (9 kGy). This indicates the potential of ionizing radiation in wastewater treatment including highly recalcitrant wastewater. However, the treatment efficiency depended upon many factors including the wastewater characteristics, as different intensities of ionizing radiation seem to effect different compositions of target wastewater. Lim and Kim [37] suggested that the carbohydrates and proteins in swine wastewater become solubilized within 20 kGy and 75 kGy irradiation dosage, making its subsequent treatment using an ion-exchange biological reactor better. Hence, a proper analysis of the wastewater constituents can reveal the reason behind a specific optimum dosage and helps in further improving the treatment efficiency.

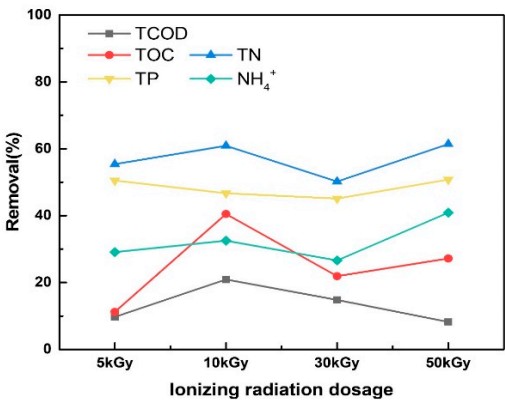

**Figure 1.** Effect of different ionizing radiation dosage on pollutants removal from landfill leachate.

The pollutant removal mechanism using ionizing radiation can be of two types: direct and indirect action on the pollutants. Often, it has been found that indirect action, i.e., the generation of hydroxyl radicals, is the governing mechanism through which pollutant degradation takes place during ionizing irradiation of pollutants containing wastewater. When water molecules absorb high-energy ionizing radiation, the reaction in Equation (1) occurs [2,26]:

$$H_2O \rightarrow [2.7]\,e_{aq}^- + [0.6]\,H + [2.8]\,HO + [0.45]\,H_2 + [0.7]\,H_2O_2 + [3.2]\,H_{aq}^+ + [0.5]\,OH_{aq}^- \tag{1}$$

The organics present in the leachate reacts with hydroxyl radical produced due to ionizing irradiation and is either mineralized to carbon dioxide and water molecules or broken down to smaller size degradation byproducts. The ammonia nitrogen removal through ionizing radiation treatment involves the following reaction scheme (Equations (2)–(7)), where through a series of reactions, $NH_4^+$ is converted to the nitrogenous end products nitrate ($NO_3^-$) and nitrogen ($N_2$).

$$2NH_4^+ \rightarrow N_2 + 8H^+ + 6e^- \tag{2}$$

$$HOCl + NH_4^+ \rightarrow NH_2Cl + H_2O + H^+ \tag{3}$$

$$HOCl + NH_2Cl \rightarrow NHCl_2 + H_2O \tag{4}$$

$$NHCl_2 + H_2O \rightarrow NOH + 2H^+ + 2Cl^- \tag{5}$$

$$NHCl_2 + NOH \rightarrow N_2 + HOCl + H^+ + Cl^- \tag{6}$$

$$4HOCl + NH_4^+ \rightarrow NO_3^- + H_2O + 6H^+ + 4Cl^- \tag{7}$$

Further reduction of nitrate to nitrite and nitrogen during ionizing radiation treatment occurs through the following reactions (Equations (8) and (9)):

$$2NO_3^- + 2H_2O + 4e^- \rightarrow 2NO_2^- + 4OH^- \tag{8}$$

$$2NO_2^- + 4H_2O + 6e^- \rightarrow N_2 + 8OH^- \tag{9}$$

From Figure 1, it can be seen that the ammonia, total nitrogen, and total phosphorus removal efficiency showed an increasing trend with the increase in the absorption dose with the highest removal at 50 kGy. On the other hand, COD and TOC removal efficiencies only improved when the irradiation dose was increased from 5 to 10 kGy. A further increase in the irradiation did not show any increase in their removal, rather there was decrease in COD and TOC removal at 30 and 50 kGy dosages compared with 10 kGy. Such contrary behavior could be attributed to the formation of a higher amount of decomposition byproducts at a high irradiation dosage due to the high amount of hydroxyl radicals generated at such dosages (30–50 kGy) [38,39]. In the case of AOP-based wastewater treatment, often it has been observed that degradation byproducts are more difficult to degrade and have a more toxic effect. This may have been the reason behind the increase in COD and TOC in the subsequent high ionization dosages. In addition, the presence of large amounts of chloride ions in the leachate promotes the formation of chlorine-related reactive substances (Cl and ClO) during ionizing irradiation, which facilitates the removal of ammonia through a series of steps (Equations (10)–(15)) but inhibits the degradation of organic matter [40–42]. An increase in the ionizing radiation dose increases the concentration of hydroxyl radicals and thus the concentration of chlorine-related reactive substances. This explains the increased removal efficiency of ammonia and total nitrogen with increasing doses. Such better performances for ammonia and total nitrogen removal using ionizing radiation have been previously reported by other authors. Lim and Kim [37] observed a maximum total nitrogen removal of 75% from swine wastewater using a 75 kGy irradiation, dosage which is relatable to the value obtained in this study. A very high $NH^{4+}$-N removal of 98.7% from spent caustic wastewater was also reported in the literature [43]; however, the initial TN value was much lower (122 mg $L^{-1}$) than the nitrogen content (11,650 mg $L^{-1}$) reported in this study. This finding is also supported by another study where an increase in the initial ammonia nitrogen concentration from 50 to 150 mg $L^{-1}$ resulted in reduction of removal efficiencies from 95 to 75%, respectively, at a 15 kGy irradiation dosage [44]. Considering these factors, the nitrogen removal efficiency obtained in this current study is superior and can provide encouraging results in a large-scale installation.

$$3Cl_2 + 2NH_4^+ \rightarrow N_2 + 8H^+ + 6Cl^- \tag{10}$$

$$ClO + NH_4^+ \rightarrow NH_2 + H^+ + HClO \tag{11}$$

$$NHCl_2 + H_2O \rightarrow NOH + 2H^+ + 2Cl^- \tag{12}$$

$$NOH + NHCl_2 \rightarrow N_2 + H_2O + 2Cl^- \tag{13}$$

$$ClO^- + NH_4^+ \rightarrow NO_3 + H_2O + 4Cl^- + 2H^+ \tag{14}$$

$$NHCl + HOCl \rightarrow NHCl_2 + OH \tag{15}$$

## 2.2. Lechate Treatment Using Coagulation Process

Figure 2 shows the optimization of the coagulation process for treatment of ionization-treated landfill leachate. The effect of different coagulants, namely $FeCl_3$, Alum, $Fe_2(SO_4)_3$,

PAC1, and PAC2, were studied under different pH ranging from 3 to 7. It is clear from this study that $FeCl_3$ showed best total COD removal (30.4%) at pH 3 (Figure 2a) and was selected as coagulant for further experiments. Another iron-based coagulant $Fe_2(SO_4)_3$ and zecoat-173 were the second best coagulants with 25% and 24.9% COD removal at pH 5 and 6, respectively. In most of the previous studies, an acidic pH is observed to be best suited for iron-based coagulants [45]. Such better performances of ferric chloride for landfill leachate treatment have been demonstrated previously by other authors as well [46,47]. Amor et al. [47] reported a 65% COD removal with 4 g $L^{-1}$ of $FeCl_3$ dosage at pH 5 in comparison to 39% removal by aluminum sulfate (2 g $L^{-1}$ at pH 6), 21% removal by calcium hydroxide (2 g $L^{-1}$ at pH 9), and 20% removal by ferrous sulfate (2 g $L^{-1}$ at pH 10 and 11). The better coagulation performance of ferric chloride is explained by the fact that ferric ions combine with hydroxyl ions to form ferric oxyhydroxide, which helps in flocculation [48,49]. Hence, this type of iron-based reagent shows a combined effect of coagulation and flocculation, resulting in a better removal efficiency of the suspended particles from the leachate.

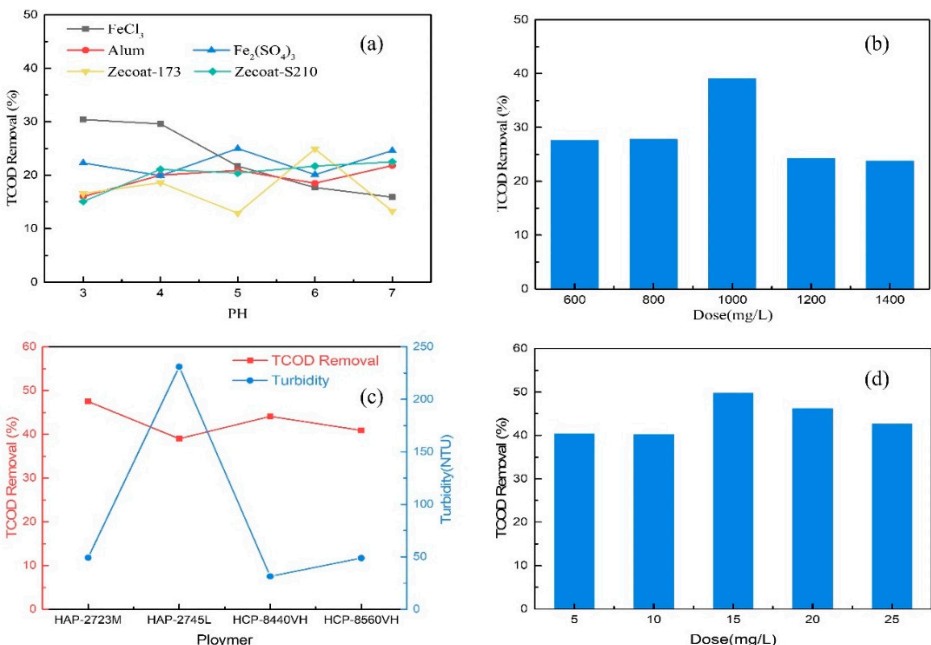

**Figure 2.** Leachate treatment using coagulation process. Effect of (**a**) pH and type of coagulant, (**b**) coagulant dosage, (**c**) type of coagulant aid, and (**d**) coagulant aid dosage.

Many factors including inorganic/organic content of the leachate, pH, stirring time, and speed are important criteria for deciding the pollutant removal performance via coagulation–flocculation. Among these many factors, pH plays a critical role as the suspended particles cannot form an agglomerate unless the pH reaches the isoelectric point of the suspended molecules in the solution [47]. Although the maximum COD removal was at pH 3, pH 4 was chosen for further studies due to the fact that the COD removal efficiency for both these pH values is similar. Considering the amount of acid required to lower the pH, a higher pH value of 4 was selected as the optimum pH in this study. Furthermore, the turbidity of the resulting solution after the coagulation process was better at pH 4 compared with pH 3 (results not shown), which provided another incentive for choosing pH 4 as the optimum pH value for the coagulation process.

Figure 2b shows the TCOD removal efficiencies at different dosages of $FeCl_3$. The TCOD removal efficiency increases with the increase in the $FeCl_3$ dosage from 600 mg $L^{-1}$ to 1000 mg $L^{-1}$, with a best TCOD removal efficiency of 39.14% at 1000 mg $L^{-1}$ $FeCl_3$ dosage. However, further increases in the $FeCl_3$ dosage to 1200 and 1400 mg $L^{-1}$ decreased the TCOD removal efficiency to even lower COD removal values than those obtained using 600–800 mg $L^{-1}$ coagulant dosages. During the coagulation

process, the ions in the coagulant can form colloidal clusters with pollutants (such as organic matter and heavy metals) present in the wastewater to be treated. However, the excess amount of coagulant can change the surface charge of the colloids, making it difficult to form colloids, leading to a decrease in the removal efficiency of pollutants [50]. This could be the reason behind low TCOD removal at a high coagulant dosage.

Later on, four different cationic and anionic polyacrylamides were investigated as coagulant aids to treat the landfill leachate. Figure 2c shows that compared to other coagulant aids, anionic polyacrylamide HAP-2723M is able to remove a significant amount of TCOD (46.5%) and obtain a low effluent turbidity (49 NTU) at the 20 mg L$^{-1}$ dosage. Further dosage optimization of HAP-2723M as a coagulant aid revealed that 15 mg L$^{-1}$ is the optimal dosage, with 49.7% TCOD removal efficiency from the landfill leachate (Figure 2d). The removal efficiency values were low for the other coagulant aid dosages. As the name suggests, coagulant aids help in the coagulation process by allowing solids to be separated more readily via gravity settling or during the filtration process [51,52]. The main role the coagulant aids serve is that they enhance the efficacy of tiny particles and flock to join together and form larger-sized flocks, which are better for their own removal [53,54]. In this study too, such an improvement due to the addition of a coagulant aid was observed as the TCOD removal improves by nearly 10% under the same experimental conditions of pH, coagulant (FeCl$_3$) concentration, and agitation. However, an optimum ratio between the coagulant and coagulant aid exists without which it is difficult for forming larger flocs. Due to this reason, a variation in pollutant removal could be observed with different coagulant aid dosages.

The effect of ionizing radiation on leachate treatment using coagulation–sedimentation was significant, particularly for COD removal. The TCOD removal increased with an increase in the ionizing radiation intensity, and this difference reached its highest at 50 kGy radiation, where the removal efficiency is 19% higher than the value obtained without any ionizing radiation treatment (Figure 3a). A similar observation was previously made by Bao et al. [55], where combined irradiation–flocculation treatment of sewage showed that the combined treatment was more successful in COD removal than either irradiation or flocculation alone. Another study showed that an additional 10% COD removal could be achieved by combined ionizing radiation and coagulation treatment compared to only irradiation [2].

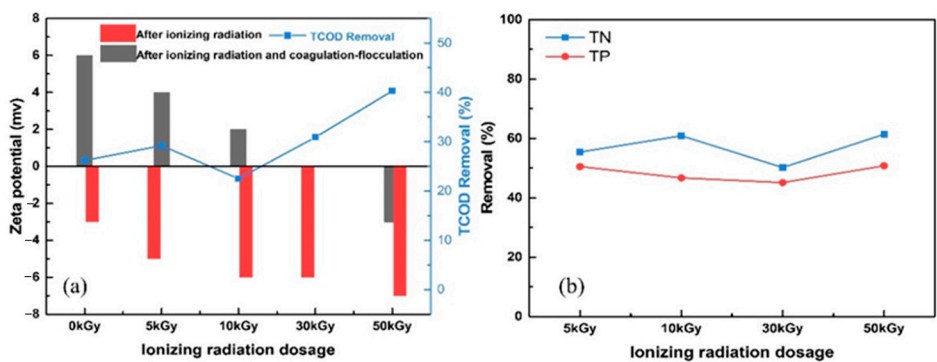

**Figure 3.** Effect of ionizing radiation on pollutant removal efficiency using coagulation–sedimentation treatment. (**a**) COD and zeta-potential, (**b**) total nitrogen and total phosphate.

The ionizing radiation causes larger size particles to be broken down into smaller particles, along with generating different radicals. This causes changes in the physical properties of the suspended particles in the raw leachate and helps them to better coagulate during the coagulation–sedimentation step to be removed more efficiently. Another important point to be noted here is that the Zeta potential value decreased proportionally to the irradiation intensity (−3 mV (0 kGy) to −7 mV (50 kGy)) (Figure 3a), which may have played a role in the better removal of organic matter during the subsequent coagulation step, even though the coagulant dosage and experimental conditions remained same. From

Figure 3b, it could also be observed that there was no significant effect of ionizing radiation on the TN and TP removal during coagulation. The reason could be attributed to the fact that nitrogen is a dissolved compound and coagulation does not have any effect on its removal [55,56]. In the case of phosphate, the removal efficiency is high for most of the experimental conditions and seems to be independent of ionizing radiation intensity.

### 2.3. Comparative Study of Coagulation–Sedimentation after Ionizing Radiation Treatment and Coagulation Process Combined with Ionizing Radiation

The comparison between different treatment sequences for ionizing radiation and coagulation showed contradictory results for the different pollutants present in the leachate. In the cases of COD and phosphate removal, the coagulation–sedimentation followed by ionizing radiation was more favorable, with 19% and 35% respective increases in their removal compared to the values obtained during simultaneous ionizing radiation treatment and coagulation (Figure 4a). However, nitrogen and ammonia removal were better with coagulation during ionizing radiation treatment. When coagulation is performed simultaneously with ionizing radiation, the addition of $FeCl_3$ as a coagulant increased the concentration of chloride ions in the leachate, which can cause a rise in the chlorine radical formation leading to a positive impact on the nitrogen removal efficiency. The interaction with chloride ions promoted the removal of ammonia via a series of reactions between hydroxyl radicals and chloride ions, as explained earlier (Equations (2)–(7)) [40–42]. However, large amounts of chloride ions inhibit organic degradation, so the organic removal was lower when the coagulation treatment was performed along with ionizing radiation. This is the reason that although nitrogen removal improved with simultaneous coagulation and ionizing radiation, the COD removal was lowered. Nevertheless, the effectiveness of ionizing radiation was again confirmed from the results shown in Figure 4b. Irradiating landfill leachate at 50 kGy radiation intensity provided an additional 3% for SCOD, 14% for total nitrogen, 25% for TP, and 27% for $NH_4^+$ removal compared to the results obtained without any ionizing treatment.

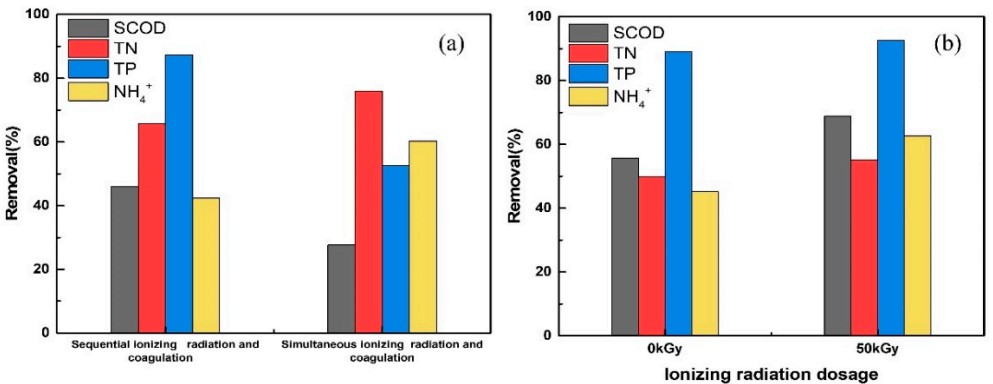

**Figure 4.** (**a**) Comparison between sequential ionizing radiation and coagulation to simultaneous ionizing radiation and coagulation treatment, and (**b**) pollutants removal during simultaneous ionizing radiation and coagulation.

### 2.4. Effects of Fenton Oxidation

Figure 5a depicts the experimental findings of the TCOD removal from leachate wastewater using the Fenton process. A maximum TCOD removal efficiency of 51.4% was obtained when the $H_2O_2$:$Fe^{2+}$ ratio was 1:1.25. The TCOD removal efficiency for other ratios, particularly those lower to this optimum value (<1:1.25), was much lower, indicating the significance of the optimum ratio between the reactant and catalyst during the Fenton process. The low iron concentration indicates that $H_2O_2$ may have failed to generate a sufficient amount of hydroxyl radicals required for reacting and degrading pollutants present in the landfill leachate. Furthermore, even at a higher $H_2O_2$:$Fe^{2+}$ ratio, the COD removal efficiency was low. This could be attributed to the fact that the presence of an

excess amount of iron causes a trapping effect on hydroxyl radicals, leading to a reduced effectiveness of hydroxyl radical oxidation of organic matter and hence the elimination rate of TCOD also reduced [17]. Another reason could be that in the case of a high $Fe^{2+}$ concentration, $H_2O_2$ rapidly oxidizes $Fe^{2+}$ to $Fe^{3+}$, causing a large amount of $H_2O_2$ to be oxidized in the process, weakening the system's catalytic activity. In the subsequent experiment, the $H_2O_2$ and $Fe^{2+}$ dosages were increased in a gradual manner (by keeping their ratio fixed at 1:1.25) to check its effect on TCOD removal from the leachate (Figure 5b). The TCOD removal efficiency increased with an increase in the $H_2O_2$ dosage with a maximum value of 62.5% at 1600 mg $L^{-1}$ $H_2O_2$ and 2000 mg $L^{-1}$ iron concentration. The amount of hydroxyl radicals produced is directly proportional to the $H_2O_2$ concentration, and hence it is no surprise that increasing the dosage improves the COD removal efficiency. The impact of ionizing radiation was again prominent even for Fenton oxidation. The pollutant removal efficiencies of Fenton oxidation under optimum condition were 60.3% TCOD, 68.9% SCOD, 55.1% TN, 92.6% TP, and 62.6% ammonia for ionizing radiation and coagulation pretreated leachate and were higher for all the parameters when compared with values obtained with samples not treated using ionizing radiation.

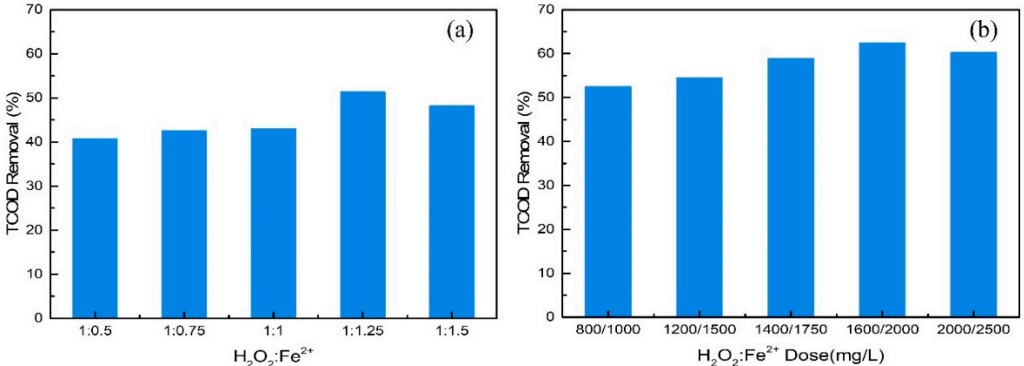

**Figure 5.** Leachate treatment using Fenton oxidation. Effect of (**a**) $H_2O_2$:$Fe^{2+}$ ratio and (**b**) $Fe^{2+}$/$H_2O_2$ dosage keeping their ratio fixed at 1:1.25.

There are a number of previous studies on Fenton treatment of landfill leachate; however, most of them focused on this method alone rather than the combined treatment approach taken in this study. Zhang et al. [57] reported nearly 62% COD removal with a $H_2O_2$ to $Fe^{2+}$ ratio of 1.5 and a pH of 2.5 within 30 min of the reaction time. Deng [58] showed 61% COD removal from a mature landfill leachate sample with a higher $H_2O_2$ to $Fe^{2+}$ ratio of 3. Another similar study found 63% COD removal, while treating mature landfill leachate with 20 mg $L^{-1}$ iron and 113 mM $H_2O_2$ dosage [47]. These pollutant removal values matched well with those observed in our current study, indicating the feasibility of this method to be implemented successfully in the industry to treat landfill leachate.

The importance of the ionizing radiation treatment is further highlighted when we compare the pollutant removal values from subsequent treatment methods (Figure 6). Here, in Table 1, two different scenarios are compared with the key difference being the addition of the ionizing radiation step. In the case of ionizing radiation being the first step in the treatment sequence, an initial 5.1% soluble COD, 61% total nitrogen, 41% ammonia nitrogen, and 51% phosphate is removed. Not only that, in the subsequent coagulation–flocculation step, the leachate treated with ionizing radiation showed much better removal (22% and 23% higher SCOD and TP) compared to the leachate not exposed to ionizing radiation. Even for the Fenton oxidation, ionizing radiation proved to be beneficial, as for almost all the different parameters, a lower effluent pollutant concentration could be achieved with this method. The coagulation–flocculation as a pretreatment to Fenton oxidation also has a positive impact on the Fenton oxidation process. The main constituent of the mature landfill leachate is humic substances [46], which along with other organic suspended particles present in the leachate are well known for their hydroxyl radical scavenging effect [47]. The

removal of such compounds through coagulation–flocculation prior to Fenton oxidation caused its better performance using an even lower optimum reactant dosage. Overall, following all the treatments, the methods including ionizing radiation treatment showed 74% SCOD, 83% TN, 63% ammonia nitrogen, and 97% phosphate removal (Figure 6, Table 1), whereas for the same leachate solution without ionizing radiation, the respective removal efficiencies were much lower at 56%, 50%, 45%, and 89% for SCOD, TN, $NH_4^+$-N, and TP.

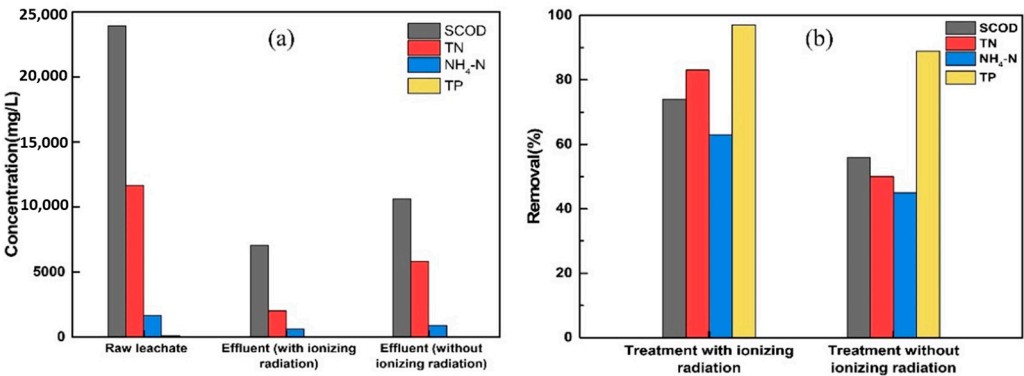

**Figure 6.** Comparative study of leachate pretreatment with and without ionizing radiation process. (**a**) Pollutant concentration and (**b**) pollutant removal efficiency for different experimental conditions.

**Table 1.** Comparative study between different treatment processes with or without the addition of ionizing radiation treatment.

| | Raw Leachate | Ionizing Radiation (50 kGy) | | Coagulation–Sedimentation | | Fenton Oxidation | | Effluent |
|---|---|---|---|---|---|---|---|---|
| | Concentration (mg L$^{-1}$) | Concentration (mg L$^{-1}$) | Removal (%) | Concentration (mg L$^{-1}$) | Removal (%) | Concentration (mg L$^{-1}$) | Removal (%) | Removal (%) |
| SCOD | 23,950 | 22,730 | 5.1 | 12,920 | 43 | 7070 | 69 | 74 |
| T-N | 11,650 | 4500 | 61 | 3990 | 11 | 2020 | 55 | 83 |
| NH$_4$-N | 1662.5 | 982.3 | 41 | 959 | 2.4 | 622 | 37 | 63 |
| T-P | 109.8 | 54 | 51 | 13.6 | 75 | 3.1 | 94 | 97 |
| | **Raw Leachate** | | | **Coagulation–Sedimentation** | | **Fenton Oxidation** | | **Effluent** |
| | Concentration (mg L$^{-1}$) | Concentration (mg L$^{-1}$) | Removal (%) | Concentration (mg L$^{-1}$) | Removal (%) | Concentration (mg L$^{-1}$) | Removal (%) | Removal (%) |
| SCOD | 23,950 | Not applicable | | 18,980 | 21 | 10,620 | 56 | 56 |
| T-N | 11,650 | | | 10,240 | 12 | 5840 | 50 | 50 |
| NH$_4$-N | 1662.5 | | | 1412.4 | 15 | 912 | 45 | 45 |
| T-P | 109.8 | | | 53 | 52 | 12 | 89 | 89 |

### 2.5. Biological Treatment as Post-Treatment to Combined Ionizing Radiation, Coagulation–Sedimentation and Fenton Oxidation Process

Biological treatment was explored in this study as a post-treatment for the landfill leachate following different physicochemical treatments such as ionizing radiation, coagulation–sedimentation, and Fenton oxidation process. Table 2 shows the total organic carbon and nitrogen concentration before and after the biological treatment conducted for a period of 24 h. The influent to the biological treatment was pre-treated via the three methods mentioned before. The results showed that TOC and TN removal was best for a low Cl$^-$ ion concentration in the wastewater. In the case of TOC removal, the best removal of 68% was for a 4% salt concentration in the leachate, whereas the maximum nitrogen removal of 73% was for a 0.5% salt concentration. The reason for such an observation is that a high salt concentration is inhibitory to the biomass growth and their normal biological activity [59]. Such an adverse effect of a high salt concentration on biological treatment is supported by previous literature as well. For instance, Ching and Redzwan [60] observed that both the pollutant removal and biomass growth were inhibited at more than 3.0% salt

concentration while treating saline fish processing wastewater using an aerobic continuous flow bioreactor system. Even in the case of anaerobic systems, biogas production and organic compound degradation were completely inhibited at a high salt concentration of 80 mS cm$^{-1}$ EC [61]. Another study by Alkaabi et al. [62] reported a significant decrease in the methane yield and organic leachate production from municipal solid waste treating anaerobic bioreactor due to the inhibition caused by a high salt concentration (3%).

**Table 2.** Effect of biological treatment on (a) TOC and (b) TN present in pre-treated landfill leachate.

| **(a)** | **Total organic carbon (TOC) concentration** | | | | | |
|---|---|---|---|---|---|---|
| Cl$^-$ concentration | 0.5% | | 4% | | 8% | |
| Irradiation intensity | 0 kGy | 50 kGy | 0 kGy | 50 kGy | 0 kGy | 50 kGy |
| Initial TOC (mg L$^{-1}$) | 140 mg L$^{-1}$ | 80 mg L$^{-1}$ | 83 mg L$^{-1}$ | 68 mg L$^{-1}$ | 58 mg L$^{-1}$ | 50 mg L$^{-1}$ |
| Final TOC (mg L$^{-1}$) | 54 mg L$^{-1}$ | 30 mg L$^{-1}$ | 64 mg L$^{-1}$ | 51 mg L$^{-1}$ | 91 mg L$^{-1}$ | 72 mg L$^{-1}$ |
| **(b)** | **Total nitrogen (TN) concentration** | | | | | |
| Cl$^-$ concentration | 0.5% | | 4% | | 8% | |
| Irradiation intensity | 0 kGy | 50 kGy | 0 kGy | 50 kGy | 0 kGy | 50 kGy |
| Initial TN (mg L$^{-1}$) | 40 mg L$^{-1}$ | 23 mg L$^{-1}$ | 66 mg L$^{-1}$ | 69 mg L$^{-1}$ | 65 mg L$^{-1}$ | 55 mg L$^{-1}$ |
| Final TN (mg L$^{-1}$) | 26 mg L$^{-1}$ | 6.2 mg L$^{-1}$ | 84 mg L$^{-1}$ | 77 mg L$^{-1}$ | 58 mg L$^{-1}$ | 57 mg L$^{-1}$ |

The final TOC value increased compared to the initial TOC value at an 8% salt concentration by almost 44%. It is well known that a high salt concentration causes osmotic pressure on bacterial cells, which can lead to cell death and cell lysis [59,62]. The organic matter present inside the cell can be released into the media under such high saline conditions causing a rise in TOC and nitrogen values [59,63]. Hence, to achieve a better treatment efficiency using a biological process, the salt concentration needs to be at a minimum. This study also clearly demonstrates the unsuitability of biological methods to treat raw landfill leachate due to the inhibitory effect of high salt concentrations present in raw leachate, which requires it to be first treated using one or more physico-chemical methods, as shown in this work. The ionizing radiation could be one such method that can be used successfully, as for all the different salt concentrations, the samples with ionizing irradiation showed a better treatment performance during the biological treatment (Table 2).

## 3. Materials and Methods

### 3.1. Collection and Characterization of Landfill Leachate

The leachate was collected from a sanitary landfill site located in the northwestern part of Republic of Korea. This landfill is a closed type and has been operated for over 30 years. Hence, the leachate obtained from this site can be characterized as mature landfill leachate. Leachate samples were collected from the retention pond and taken to the laboratory for storage as soon as possible. To avoid biological or chemical changes, samples are kept in a refrigerator at 4 °C. The leachate was characterized using the standard method [64], and the main leachate characteristics were as follows; pH: 9.74; conductivity: 337,200 μs cm$^{-1}$; suspended solids (SS): 227,280 mg L$^{-1}$; total chemical oxygen demand (TCOD) concentration: 26,940 mg L$^{-1}$; soluble chemical oxygen demand (SCOD) concentration: 23,950 mg L$^{-1}$; total organic content (TOC) concentration: 4697 mg L$^{-1}$; ammonia nitrogen: 6117 mg L$^{-1}$; total nitrogen (TN): 116,506,117 mg L$^{-1}$; total phosphorus (TP): 109.8 mg L$^{-1}$; chloride (Cl$^-$) concentration: 89,673 mg L$^{-1}$; and turbidity: 13.05 NTU.

### 3.2. Leachate Treatment Using Ionizing Radiation

The reactor for ionizing radiation treatment was manufactured using a stainless-steel plate made of SUS316. The diameter and height of the reactor were 250 mm and 400 mm, respectively, and a circular diffuser was installed at the bottom to supply compressed air. A total of 600 mL of leachate was added to the reactor during electron beam irradiation and was performed while continuously supplying compressed air to the reactor. As compressed

air was supplied to the reactor, bubbles were generated in the leachate inside the reactor creating conditions for efficient electron beam irradiation. A 10 MeV electron beam from the Rhodotron TT200 gas pedal at the Korea Atomic Energy Research Institute was used for the irradiation step, outputting 5 and 10 MeV electron beam lines with a maximum power of 100 kW. The absorbed dose at the location of the samples was measured using a CTA film dosimeter. All experiments were carried out at room temperature. In this study, different irradiation doses (5–50 kGy) were used to investigate the effect of irradiation dose on contaminant removal.

### 3.3. Leachate Treatment Using Coagulation

The coagulation test was performed using a jar test apparatus with 1 L volume cylindrical beakers. The experimental setup had impellers affixed to rectangular blades to stir the liquid during the reaction. The reaction volume was 400 mL. Initially, the effect of four different coagulants, namely ferric chloride ($FeCl_3$), alum, ferrous sulfate ($Fe_2(SO_4)_3$), poly aluminum chlorides (PAC1 and PAC2), at different pH ranging from 3 to 7 were studied. The solution pH was adjusted using 1 M $H_2SO_4$ or 1 M NaOH. After, the coagulant was then added to the leachate samples, the solution was mixed in two phases: first, rapid mixing at 150 rpm for 5 min and then, slow mixing at 30 rpm for 20 min. After the mixing, the reaction mixture was left alone for 60 min for flocs to settle. Since $FeCl_3$ showed a better result compared to other coagulants, it was used in further experiment where its optimum dosage was found out by adding it within the range of 600–1400 mg $L^{-1}$. Cationic and anionic polyacrylamide (HAP-2723M, HAP-2745L, HCP-8440VH, and HCP-8560VH) were used as coagulant aids to evaluate their effectiveness to reduce TCOD and turbidity of treated leachate. Further optimization of coagulant aid dosage was performed at 5–25 mg $L^{-1}$ range with HAP-2723M, as it showed the best removal result.

### 3.4. Simultaneous Ionizing Radiation with Coagulation–Sedimentation

In order to study the effectiveness of ionizing irradiation together with coagulation treatment, a sperate experiment was conducted, during which optimized dosage and experimental conditions obtained from the previous section (Section 2.3) were used. The experiment used $FeCl_3$ as the coagulant (1000 mg $L^{-1}$) together with an anionic coagulant aid (HAP-2723M, 15 mg $L^{-1}$) at pH 4. The experimental conditions were rapid mixing at 150 rpm for 5 min, followed by slow mixing at 30 rpm for 20 min, and settling time of 60 min (without any mixing).

### 3.5. Leachate Treatment Using Fenton Oxidation

Similar to the coagulation experiment, a jar test apparatus was used for Fenton oxidation as well. A total of 400 mL of leachate solution following coagulation was taken in 1 L volume beaker and a fixed amount of freshly prepared $FeSO_4$ and 35% $H_2O_2$ were added to the wastewater. The pH of the solution was adjusted to 4 using 1M $H_2SO_4$. The oxidation experiment was carried out by mixing the reactant and catalyst at 150 rpm for 60 min, then followed by adjusting the solution pH to 8, and again mixed rapidly at 150 rpm for 5 min. This increase in pH to 8 stops the Fenton reaction by oxidizing residual $H_2O_2$ present in the solution [65]. Finally, the solution was mixed at slower pace of 30 rpm for 20 min., before being allowed to settle for 60 min. Again, the optimal conditions were determined using single-factor experiments. For optimizing the iron dosage during Fenton oxidation, hydrogen peroxide concentration was kept fixed at 1000 mg $L^{-1}$ (COD:$H_2O_2$ = 7.1:1) and the iron concentration was varied to obtain an $H_2O_2$ to $Fe^{2+}$ of 1:0.5, 1:0.75, 1:1, 1:1.25, and 1:1.5. Later, at the optimal $H_2O_2$:$Fe^{2+}$ ratio of 1:1.25, the dosage of these two reactants were varied from initial 800:100 to 1200:1500, 1400:1750, 1600:2000, and 2000:2500 mg $L^{-1}$ to determine the optimum condition for leachate treatment. The respective COD to $H_2O_2$ ratios for these dosages were as follows: 8.8:1, 5.9:1, 5:1, 4.4:1, and 3.5:1.

### 3.6. Biological Treatment of Pretreated Landfill Leachate

Biological treatment of landfill leachate treated with ionizing radiation, coagulation, and Fenton oxidation was performed using activated sludge in a 2 L volume beaker. For experimental purposes, 50 mL of leachate solution was added to 950 mL of activated sludge to make up the reaction volume to 1 L. Since the leachate used in this study has high salt content, the effect of salt concentration (i.e., chloride ion) on COD and nitrogen removal by activated sludge was examined. The original chloride ion concentration was around 9% in the raw leachate, which was reduced to 0.5% in leachate following three pretreatments applied to it. Due to this reason, three chloride concentrations, i.e., 0.5%, 4%, and 8%, were selected to evaluate the effect of salt concentration on feasibility of biological treatment. The desired salt concentration in leachate was achieved by adding desired amount of NaCl. The beaker containing leachate solution and activated sludge was continuously supplied with air during the experiment to ensure the DO level was more than 3. The pH value was controlled at 7.0–7.3 to provide ideal conditions for biological treatment. The biological reaction continued for a duration of 24 h. Following the reaction, samples were filtered using a GFC filter (Merck Millipore, Darmstadt, Germany) prior to their analysis.

### 3.7. Analytical Methods

COD was measured calorimetrically using a DR/4000U spectro-photometer (HACH Company, Loveland, CO, USA) using standard closed reflux method. The COD analysis was performed with samples that are diluted 5 times to avoid interference due to $Cl^-$ ions present in the samples. The pH values were determined using pH meter (pH Meter, model ST300, Ohaus, Parsippany, NJ, USA), and turbidity values were determined using turbidimeter (2100N, Turbidimeter, HACH, Loveland, CO, USA). Water samples were passed through 0.45 μm acetate membranes for TOC analysis, and TOC was determined using a total organic carbon analyzer (TOC, TOC-5000A, Shimadzu, Kyoto, Japan). Anions and cations were determined using Ion Chromatography-Mass Spectrometer, Metrohm (Herisau, Switzerland).

## 4. Conclusions

This study demonstrated the feasibility of a combined treatment approach involving ionizing radiation, coagulation, Fenton oxidation, and biological processes for landfill leachate. A maximum of 27% of TOC, 61% of nitrogen, and 51% of phosphorus removal efficiency was achieved along with a very high (93%) $NO_3^-$-N removal at 50 kGy ionizing radiation. The reduction of Zeta potential from $-3$ to $-7$ mV via ionizing irradiation appears to increase the efficiency of downstream coagulation/sedimentation treatment. Among the different coagulants studied, $FeCl_3$ showed the best pollutant removal properties. Further, the optimization of the coagulation process was carried out, and under the optimum condition, the total COD removal improved to 45.1%. Another AOP-based treatment method, Fenton oxidation, was explored for treating leachate. The pollutant removal increased to 62.5% TCOD, 55.10% TN, 62.6% $NH_4^+$, and 92.6% TP with the addition of Fenton oxidation along with the treatment chain. Finally, biological treatment was examined as the post-treatment followed by the other three treatment methods, which showed an additional 25% TCOD and 73% of nitrogen removal. However, the effect of chloride ions on the biological treatment was significant, and an effective treatment could only be achieved at low salt (0.5–4%) concentrations. Most significantly, the importance of ionizing radiation treatment on the subsequent treatments including biological treatment was highlighted in this study.

**Author Contributions:** Conceptualization, C.-M.C. and M.-J.L.; methodology, C.-M.C. and M.-J.L.; formal analysis, S.L. and G.-Y.L.; Resources, M.-J.L. and B.-C.L.; data curation, S.L. and A.S.; writing—review and editing, S.L., A.S. and C.-M.C.; supervision C.-M.C. All authors have read and agreed to the published version of the manuscript.

**Funding:** This work was funded by the Technology development Program (No. S3276954), funded by the Ministry of SMEs and Startups (MSS, Republic of Korea).

**Data Availability Statement:** The data is available upon reasonable request to the authors.

**Acknowledgments:** The authors would like to thank the Uniscan, Co., Ltd. for providing the equipment and technical support for the experiments.

**Conflicts of Interest:** The authors declare no conflict of interest.

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
