# Peer review of "Synergistic Effects of Ionizing Radiation Process in the Integrated Coagulation–Sedimentation, Fenton Oxidation, and Biological Process for Treatment of Leachate Wastewater"

_catalysts, doi:10.3390/catal13101376_

Round 1

Reviewer 1 Report

Article entitled “Synergistic effects of ionizing radiation process in the integrated coagulation-sedimentation, Fenton oxidation and biological process for treatment of leachate wastewater” written by Sha-Liu1, Arindam Sinharoy, Ga-Young Lee and Chong-Min Chung submitted to Catalysts journal as a draft no 2643011 deals with an important issue of leachate wastewater treatment (GW) by ionizing radiation process IRP in the integrated coagulation-sedimentation (C/S), Fenton process (FP) and biological process. However, my recommendation to publish the article depends on obtaining answers to the following questions.

1. The authors wrote: “The leachate was characterized using standard method [31], …” (line 90), “COD was measured calorimetrically using a DR/4000U spectro-photometer (HACH Company, USA) using standard closed reflux method.” (line 159). Knowing the concentration range COD of the HACH methods and its limitations in use for  chloride content, please describe in detail the modifications to the COD determination method. If the authors only write that they determined in a diluted sample, I will have further questions about the concentrations (Cl-) in the diluted sample and the determination error.

2. C/S tests were carried out in the pH range 3-7, when it is known that coagulation using iron based coagulats has two optimal pH ranges for the process, and C/S with aluminum based coagulants can occur most effectively even slightly above pH 7?

3. Based on the description in the section “Leachate treatment using Fenton oxidation” ( “Similar to the coagulation experiment, jar test apparatus was used for Fenton oxidation as well. 400 mL of leachate solution following coagulation was taken in 1 L volume beaker and a fixed amount of freshly prepared FeSO4 and 35% H2O2 was added to the wastewater, it must be concluded that the description is wrong and incomplete, or the experiment was performed incorrectly. Was the pH adjusted after coagulation? If so, what happened in the sample. Was it the sample clear? Was there any turbidity? Did a occur sludge? If so, why?

4. Is there any residual hydrogen peroxide left in the sample after the FP process? This is extremely important when marking COD. Raising the pH to 8 does not guarantee the breakdown of remaining hydrogen peroxide. Therefore, the pH is most often adjusted to 9 (hydrogen peroxide decomposition and secondary coagulation).

5. In the FP process, the proportions of the reagents used are important. However, their doses should be determined in relation to the amount of contaminants in the sample expressed in terms of COD or TOC. Why was this not taken into account when conducting the research? Was the given dose sufficient? “For optimizing the iron dosage during Fenton oxidation, hydrogen peroxide concentration was kept constant at 1000 mg L-1 …”

6. In the FP process, the proportions of the reagents used are important. However, their doses should be determined in relation to the amount of contaminants in the sample expressed in terms of COD or TOC. Why was this not taken into account when conducting the research? Was the given dose sufficient? “For optimizing the iron dosage during Fenton oxidation, hydrogen peroxide concentration was kept constant at 1000 mg L-1 …”

7.   Is it possible to obtain a negative value for the removal of contaminants of the process efficiency? (Table 2.)

8. The discussion of the obtained results is insufficient and there is no reference to the results obtained by other researchers.  There are many papers on this topic.

9. The article does not provide information whether the landfill is open or closed. The time that has passed since the closure of the landfill is crucial because the composition of the leachate changes over time. At the beginning, there is a significant amount of volatile fatty acids in the leachate, the share of which decreases over time, and the amount of humic substances increases. Over time, the susceptibility of leachates to treatment using biological methods decreases.

Finally, research results should not be presented in the materials and methods section.

Author Response

  1. The authors wrote: “The leachate was characterized using standard method [31], …” (line 90), “COD was measured calorimetrically using a DR/4000U spectro-photometer (HACH Company, USA) using standard closed reflux method.” (line 159). Knowing the concentration range COD of the HACH methods and its limitations in use for chloride content, please describe in detail the modifications to the COD determination method. If the authors only write that they determined in a diluted sample, I will have further questions about the concentrations (Cl-) in the diluted sample and the determination error.

Response: The Reviewer is correct in pointing out impact of Cl- ions on COD estimation. The HACH COD kit with chloride concentrations up to 20,000 mg/L Cl- used in this study was of the caliber to handle high salt concentration. Furthermore, the COD was performed using diluted sample (5 times dilution) at which the amount of Cl- ions was below 18,000 mg L-1. COD estimation of samples containing such Cl- ions have been performed routinely including in industry. Hence, according to us the COD results are valid and didn’t significantly impact by salt concentration. 

Information about this dilution is suitably mentioned in the revised manuscript (Page 12, line no. 469-471)

  1. C/S tests were carried out in the pH range 3-7, when it is known that coagulation using iron based coagulants has two optimal pH ranges for the process, and C/S with aluminum based coagulants can occur most effectively even slightly above pH 7?

Response: It is correct that the optimum pH range for coagulation using iron-based coagulants exists within pH 4-5.5. However, as the study was to compare the performance of a number of coagulants including iron and aluminum-based coagulants, conducting experiments involving a wide pH range not only provide a better clarity on the optimum pH but also illustrate the performance of each coagulant for all the individual pH values. Hence, it was decided to conduct the study using a wider range of pH 3-7.  

  1. Based on the description in the section “Leachate treatment using Fenton oxidation” (“Similar to the coagulation experiment, jar test apparatus was used for Fenton oxidation as well. 400 mL of leachate solution following coagulation was taken in 1 L volume beaker and a fixed amount of freshly prepared FeSO4 and 35% H2O2 was added to the wastewater, it must be concluded that the description is wrong and incomplete, or the experiment was performed incorrectly. Was the pH adjusted after coagulation? If so, what happened in the sample. Was it the sample clear? Was there any turbidity? Did a occur sludge? If so, why?

Response: The pH adjustment was carried out as part of the Fenton oxidation experiment as pH plays a very important role during Fenton oxidation process. However, the effect of this pH adjustment on solution turbidity was not measured. Hence, no specific information on this matter could be provided.

As per the Reviewer’s suggestion, information on pH adjustment following coagulation experiment is included in the revised manuscript as follows:

“The pH of the solution was adjusted to 4 using 1M H2SO4” (Page 11, line no. 438)

  1. Is there any residual hydrogen peroxide left in the sample after the FP process? This is extremely important when marking COD. Raising the pH to 8 does not guarantee the breakdown of remaining hydrogen peroxide. Therefore, the pH is most often adjusted to 9 (hydrogen peroxide decomposition and secondary coagulation).

Response: There are many previous papers which have mentioned that raising solution pH to 8 can oxidize residual hydrogen peroxide and stop Fenton’s oxidation. According to which the pH 8 was used in this study after Fenton oxidation experiment. If at all any residual H2O2 remains after pH 8, we don’t think its impact on the COD value will be significant enough to alter the entire results.

Further suitable reference is included in this section for better clarity. (Page 11, line no. 440-442)

  1. In the FP process, the proportions of the reagents used are important. However, their doses should be determined in relation to the amount of contaminants in the sample expressed in terms of COD or TOC. Why was this not taken into account when conducting the research? Was the given dose sufficient? “For optimizing the iron dosage during Fenton oxidation, hydrogen peroxide concentration was kept constant at 1000 mg L-1 …”

Response: The iron to hydrogen peroxide ratio plays most important role in hydroxyl radical generation. In order to optimize the ratio initially the H2O2 concentration was kept constant and iron concentration increased. Further optimization of the dosage was carried out in the subsequent experiment where both their dosages were increased by keeping the ratio fixed. This way the optimization process for Fenton oxidation was carried out. The dosage ratio optimization with respect to COD or TOC is not a common practice in the literature, hence was not chosen as one of the objectives. However, for better clarity on this matter the resulting ratios of H2O2 to COD during optimization experiment are mentioned in the methodology section as shown below. 

“The respective COD to H2O2 ratios for these dosages were as follows: 8.8:1, 5.9:1, 5:1, 4.4:1, 3.5:1.” (Page 11, line no. 449-450)

  1. In the FP process, the proportions of the reagents used are important. However, their doses should be determined in relation to the amount of contaminants in the sample expressed in terms of COD or TOC. Why was this not taken into account when conducting the research? Was the given dose sufficient? “For optimizing the iron dosage during Fenton oxidation, hydrogen peroxide concentration was kept constant at 1000 mg L-1 …”

Response: This comment is identical to the previous comment (No. 5). Kindly see the response to the previous comment in this regard.

  1. Is it possible to obtain a negative value for the removal of contaminants of the process efficiency? (Table 2.)

Response: We understand that it could be misleading to have a negative removal efficiency value, hence, removal % values were removed from the Table 2 and only the initial and final pollutant concentrations were mentioned in the revised manuscript.

  1. The discussion of the obtained results is insufficient and there is no reference to the results obtained by other researchers. There are many papers on this topic.

Response: Further discussion on each of the section by comparing results obtained by previous studies are now included in the revised manuscript. Please see the following for more information:

Section 2.1:

“Although, there are not many previous works focusing on landfill leachate treatment using ionizing radiation, the results obtained in this work are comparable with that observed in the limited previous studies available. For example, Bae et al. [36] reported a 20.7% removal of dissolved organic carbon using 15 kGy irradiation which is relatable to the TOC removal value obtained in this current work. Another study reported a maximum of 45.3% COD removal from leachate at an irradiation dose of 4 kGy [2]. The low initial COD concentration of the leachate (950–983.3 mg L-1) was the reason behind better performance at low irradiation intensity compared to this current study. The main objective of ionizing radiation treatment is not direct pollutant removal rather make the pollutants more susceptible for subsequent treatment including biological treatment as evidenced by reduction in less biodegradable humic substances by 33.6% in landfill leachate [36]. This improvement in treatment efficiency was not only reported for landfill leachate but also for other type of wastewater such as textile industry [37], papermill [38] and swine wastewater [39]. A study reported that biodegradability of textile and dye wastewater increased by as high as 224% due to ionizing radiation treatment (9 kGy). This indicates the potential of ionizing radiation in wastewater treatment including highly recalcitrant wastewater. However, the treatment efficiency in depended upon many factors including the wastewater characteristics as different intensities of ionizing radiation seem to effect different compositions of target wastewater. Lim and Kim [39] suggested that the carbohydrates and proteins in swine wastewater get solubilized within 20 kGy and 75 kGy irradiation dosage making its treatment by a subsequent ion-exchange biological reactor better. Hence, a proper analysis of the wastewater constituents can reveal the reason behind a specific optimum dosage and helps in further improving the treatment efficiency.” (Page 2-3, line no. 89-112)

“Lim and Kim [39] observed a maximum total nitrogen removal of 75% from swine wastewater using 75 kGy irradiation dosage which is relatable to the value obtained in this study. A very high NH4+-N removal of 98.7% from spent caustic wastewater was also reported in literature [45], however, the initial TN value was much lower (122 mg L-1) than the nitrogen content (11,650 mg L-1) reported in this current study. This finding is also supported by another study where increase in initial ammonia nitrogen concentration from 50 to 150 mg L-1 resulted in reduction of removal efficiencies from 95 and 75 %, respectively, at 15 kGy irradiation dosage [46]. Considering these factors, the nitrogen removal efficiency obtained in this current study is superior and can provide encouraging results in large-scale installation.” (Page 4, line no. 157-166)

Section 2.2:

“Such better performance of ferric chloride for landfill leachate treatment has been demonstrated previously by other authors as well [48, 49]. Amor et al. [49] reported 65% COD removal with 4 g L-1 of FeCl3 dosage at pH 5 in comparison to 39% removal by aluminum sulfate (2 g L-1 at pH 6), 21% removal by calcium hydroxide (2 g L-1 at pH 9) and 20% removal by ferrous sulfate (2 g L-1 at pH 10 & 11). The better coagulation performance by ferric chloride is explained by the fact that ferric ions combining with hydroxyl ions form ferric oxyhydroxide which helps in flocculation [50, 51]. Hence, this type of iron-based reagents shows combined effect of coagulation and flocculation, resulting in better removal efficiency of the suspended particles form leachate.

Many factors including inorganic/organic content of the leachate, pH, stirring time and speed are important criteria for deciding the pollutant removal performance by coagulation-flocculation. Among these many factors, pH plays a critical role as the suspended particles cannot form agglomerate unless the pH reaches isoelectric point of the suspended molecules in the solution [49].” (Page 5, line no. 181-197)

“Similar observation was previously made by Bao et al. [57] where combined irradiation-flocculation treatment of sewage showed that the combined treatment was more successful in COD removal than either irradiation or flocculation alone. Another study showed that an additional 10% COD removal could be achieved by combined ionizing radiation and coagulation treatment compared to only irradiation [2].” (Page 6, line no. 235-240)

Section 2.4:

“There are a number of previous studies on Fenton treatment of landfill leachate, however, most of them focused this method alone rather than a combined treatment approach taken in this current study. Zhang et al. [59] reported nearly 62% COD removal with H2O2 to Fe2+ ratio of 1.5 and a pH of 2.5 within 30 min of reaction time. Deng [60] showed 61% COD removal from a mature landfill leachate sample with a higher H2O2 to Fe2+ ratio of 3. Another similar study found 63% COD removal while treating mature landfill leachate with 20 mg L−1 iron and 113 mM H2O2 dosage [49]. These pollutant removal values matched well with those observed in our current study indicating feasibility of this method to be implemented successfully in the industry to treat landfill leachate.” (Page 8, line no. 309-317)

“The coagulation-flocculation as pretreatment to Fenton oxidation also has positive impact on the Fenton oxidation process. The main constituent of mature landfill leachate is humic substances [48], which along with other organic suspended particles present in leachate are well known for hydroxyl radical scavenging effect [49]. Removal of such compounds through coagulation-flocculation prior to Fenton oxidation caused its better performance using even lower optimum reactant dosage.” (Page 8, line no. 331-336)

Section 2.5:

“Such adverse effect of high salt concentration on biological treatment is supported by previous literature as well. For instance, Ching and Redzwan [62] observed that both pollutant removal and biomass growth get inhibited at more than 3.0% salt concentration while treating saline fish processing wastewater using an aerobic continuous flow bioreactor system. Even in case of anaerobic systems biogas production and organic compound degradation completely inhibited at high salt concentration of 80 mS cm−1 EC [63]. Another study by Alkaabi et al. [64] reported significant decrease in methane yield and organic leachate production from municipal solid waste treating anaerobic bioreactor due to inhibition caused by high salt concentration (3%).” (Page 9-10, line no. 358-367)

  1. The article does not provide information whether the landfill is open or closed. The time that has passed since the closure of the landfill is crucial because the composition of the leachate changes over time. At the beginning, there is a significant amount of volatile fatty acids in the leachate, the share of which decreases over time, and the amount of humic substances increases. Over time, the susceptibility of leachates to treatment using biological methods decreases.

Response: The landfill was of closed type and over 30 years old. As rightly mentioned by the Reviewer that mature landfill leachate contains more humic substances than volatile fatty acids. However, neither of these compounds are estimated in our leachate samples, hence, no further information on this could be provided. Regarding biodegradability of leachate, we agree with the Reviewer, that mature landfill leachate is difficult to degrade. Hence, our study focused on finding treatment solution through a chain of physico-chemical treatment methods and showed that biodegradability could be improved though such treatment scheme.  

The following statement on the type and age of the landfill is added in the revised manuscript as per the suggestion of the Reviewer:

“The landfill was a closed type and is being operated for over 30 years. Hence, the leachate obtained from this site can be characterized as mature landfill.” (Page 10, line no. 385-386)

  1. Finally, research results should not be presented in the materials and methods section.

Response: There are no results presented in materials and methods section. In few places, the optimized condition from previous experiment is mentioned in order to provide details about experimental condition and to provide explanation for why such conditions was chosen. 

Reviewer 2 Report

The manuscript entitled “Synergistic effects of ionizing radiation process in the integrated coagulation-sedimentation, Fenton oxidation and biological process for treatment of leachate wastewater” provides some good results. Therefore, the current manuscript could be considered for publication, but after going through a minor revision.      

1.      The authors should use keywords that are different from those used in the main title.

2.      Error bars are missing in all the figures.

3.      Quality of all of the figures must be improved.

4.      There should be more comparisons with previously published articles that stated the application of AOPs. Some examples of the recent articles that could be useful for drawing such comparisons and enriching the manuscript are:

https://doi.org/10.1016/j.scitotenv.2019.134425

https://doi.org/10.1007/s13399-023-04893-4

https://doi.org/10.1016/j.scitotenv.2018.07.415

https://doi.org/10.1021/acsomega.1c07209

https://doi.org/10.1016/j.apcatb.2018.03.016  

English language seems to be fine, but very few grammatical mistakes should be corrected. 

Author Response

  1. The authors should use keywords that are different from those used in the main title.

Response: Keywords are chosen based on their importance and how best they represent the article. Due to this reason, no further addition to the existing keywords are carried out.  

  1. Error bars are missing in all the figures.

Response: The experiments were conducted only once due to constraints with amount of available leachate sample. Hence, it is not possible to include error bars in figures at this moment.

  1. Quality of all of the figures must be improved.

Response: The figures are improved including change in colour and font size so that they are better visible when published.

  1. There should be more comparisons with previously published articles that stated the application of AOPs. Some examples of the recent articles that could be useful for drawing such comparisons and enriching the manuscript are:

https://doi.org/10.1016/j.scitotenv.2019.134425

https://doi.org/10.1007/s13399-023-04893-4

https://doi.org/10.1016/j.scitotenv.2018.07.415

https://doi.org/10.1021/acsomega.1c07209

https://doi.org/10.1016/j.apcatb.2018.03.016 

Response: The suggested references are included in the revised manuscript. Further, many important references are added as part of the discussion in different sections.  

Reviewer 3 Report

This study compared the removal efficiency of Ionizing radiation, Fenton oxidation, and coagulation methods for treating leachate wastewater. The study compares the efficiency of physicochemical removal methods to biological treatment methods in high/low salt concentrations.  This work contains some new results and could be considered for publication. However, the authors should revise their manuscript before acceptance for publication according to the following comments:

  1. The abstract section must be informative and must compare the performance with standard materials.
  2. Throughout the text, you never define some of the acronyms, Before using abbreviations, it is necessary to provide their full forms during their first occurrence.
  3. The author addressed most of the findings without citing any references! Many parts require references. Recent literature should be updated. Please encourage the author to further discuss the acquired results by citing similar articles that demonstrated the same results;
  4. All the figures are poor and must be revised and discussed accordingly in the manuscript. The table format must match with journal style. The author must pay attention to journal templates.
  5. Can the author describe a mechanism for this study?
  6. A comparison of physicochemical treatment methods used in this study should be made with those of the related samples reported in the literature in terms of the method used and removal efficiencies.
  7. There are some inappropriate English words or expressions in the manuscript. The authors should carefully polish the English of the whole manuscript.
  1. There are some inappropriate English words or expressions in the manuscript. The authors should carefully polish the English of the whole manuscript.

Author Response

  1. The abstract section must be informative and must compare the performance with standard materials.

Response: More information on results is included in the abstract, however, due to word limit suggested by the journal we could not increase its content beyond a fixed limit.  

  1. Throughout the text, you never define some of the acronyms, before using abbreviations, it is necessary to provide their full forms during their first occurrence.

Response: All abbreviations are defined in text whichever place they are mentioned first.

  1. The author addressed most of the findings without citing any references! Many parts require references. Recent literature should be updated. Please encourage the author to further discuss the acquired results by citing similar articles that demonstrated the same results.

Response: The discussion is improved by citing references wherever applicable. Further more recent references are included. Comparative discussion as per the Reviewer’s suggestion is also undertaken for different sections.

Please see detailed response to previous comment (Sl. No. 8, #Reviewer 1) for more information regarding this point.

  1. All the figures are poor and must be revised and discussed accordingly in the manuscript. The table format must match with journal style. The author must pay attention to journal templates.

Response: All the figures are improved for their better visibility. Further changes are made in the tables as well to match with the journal format.    

  1. Can the author describe a mechanism for this study?

Response: The mechanism is described with reaction scheme in different sections throughout the manuscript. The exact mechanism for ionizing radiation is not clear eve in the literature. However, all the different aspects of ionizing radiation mediated removal of COD and different nitrogen species is mentioned. 

  1. A comparison of physicochemical treatment methods used in this study should be made with those of the related samples reported in the literature in terms of the method used and removal efficiencies.

Response: Comparison of previous physicochemical treatments with the results obtained in this study is now added in the revised manuscript. Please see the detailed response to a previous comment (Sl. No. 8, #Reviewer 1) for more information.

  1. There are some inappropriate English words or expressions in the manuscript. The authors should carefully polish the English of the whole manuscript.

Response: The manuscript is thoroughly checked and all such mistakes are corrected in the revised manuscript. 

Reviewer 4 Report

1. A maximum 27% of TOC, 61% of nitrogen, and 51% of phosphorus removal efficiency was achieved along with a very high (93%) NO3--N removal at 50 kGy ionizing radiation. It is not a high result for 27% of TOC.

2. The ammonia nitrogen removal through ionizing radiation treatment involves following reaction scheme (Eq. 2-7), where through a series of reactions NH4+ gets converted to nitrogenous end products nitrate (NO3−) and nitrogen (N2). The mechanism requires careful analysis and comparative literature discussion.

3. In Figure 2, The pH has a significant impact on catalytic activity, and the mechanism and discussion are provided.

4. The decomposition of pollutants is also closely related to the surface charge of catalysts, and an explanation is needed.

good

Author Response

  1. A maximum 27% of TOC, 61% of nitrogen, and 51% of phosphorus removal efficiency was achieved along with a very high (93%) NO3--N removal at 50 kGy ionizing radiation. It is not a high result for 27% of TOC.

Response: Here, high removal is not mentioned for TOC, rather only for NO3—N which is 93%. Hope this clarify the doubt.

  1. The ammonia nitrogen removal through ionizing radiation treatment involves following reaction scheme (Eq. 2-7), where through a series of reactions NH4+ gets converted to nitrogenous end products nitrate (NO3−) and nitrogen (N2). The mechanism requires careful analysis and comparative literature discussion.

Response: More comparative discussion is included for ammonia removal during ionizing radiation as follows:

“Lim and Kim [39] observed a maximum total nitrogen removal of 75% from swine wastewater using 75 kGy irradiation dosage which is relatable to the value obtained in this study. A very high NH4+-N removal of 98.7% from spent caustic wastewater was also reported in literature [45], however, the initial TN value was much lower (122 mg L-1) than the nitrogen content (11,650 mg L-1) reported in this current study. This finding is also supported by another study where increase in initial ammonia nitrogen concentration from 50 to 150 mg L-1 resulted in reduction of removal efficiencies from 95 and 75 %, respectively, at 15 kGy irradiation dosage [46]. Considering these factors, the nitrogen removal efficiency obtained in this current study is superior and can provide encouraging results in large-scale installation.” (Page 4, line no. 157-166)

  1. In Figure 2, The pH has a significant impact on catalytic activity, and the mechanism and discussion are provided.

Response: In Fig. 2, the effect of pH was only studied for coagulation experiment. The effect of pH on catalytic activity was not performed as only the reactant dosage was considered to be significant parameter effecting process performance. Hence, no further discussion no this aspect could be included in the revised manuscript.  

  1. The decomposition of pollutants is also closely related to the surface charge of catalysts, and an explanation is needed.

Response: The surface charge of the catalyst was not relevant in this study and hence no experiment related to this was performed. The surface charge measurement (as zeta potential) was only performed during coagulation experiment as it has great impact on the coagulation process. Furthermore, it was also observed that ionizing radiation could improve subsequent coagulation experiment by altering the surface charge of suspended particles. This aspect is discussed in the manuscript.

Round 2

Reviewer 1 Report

Thank you to the authors for their explanations. I consider them sufficient. However, I do not agree with explanations 5 and 6. They are not true. If it were as mentioned by the authors, why not use ten times smaller doses of Fe+2 and H2O2.

Reviewer 3 Report

The author has successfully addressed all the comments. Moreover, author has made significant changes in the manuscript to justify the reviewer's requirement. I feel the revised manuscript is now well readable and eligible to be published in this journal.